# REAR: Scalable Test-time Preference Realignment through Reward Decomposition

## Abstract

Aligning large language models (LLMs) with diverse user preferences is a critical yet challenging task. While post-training methods can adapt models to specific needs, they often require costly data curation and additional training. Test-time scaling (TTS) presents an efficient, training-free alternative, but its application has been largely limited to verifiable domains like mathematics and coding, where response correctness is easily judged. To extend TTS to the domain of preference alignment, we introduce a novel framework that models the task as a realignment problem, as the base model often fails to sufficiently align with the preference. Our key insight is to decompose the underlying reward function into two components: one related to the question and the other to user preference. This allows us to derive a REAlignment Reward (REAR) that selectively rescales the preference-related reward while preserving the question-related reward. We show that REAR can be formulated as a linear combination of policy probabilities, making it computationally efficient and easy to integrate with existing TTS algorithms like best-of-N sampling and tree-search algorithms. Experiments on various preference alignment and role-playing benchmarks demonstrate that TTS with REAR enables scalable and effective test-time realignment with superior performance.

## 1 Introduction

The remarkable success of Large Language Models (LLMs) in aligning with human preferences is largely attributed to techniques such as Reinforcement Learning from Human Feedback (RLHF) (Ouyang et al., 2022; Bai et al., 2022; Rafailov et al., 2023; Guo et al., 2025). This alignment enables a wide range of applications, from personalized assistants (OpenAI, 2023; Chen et al., 2024; Cui et al., 2024) to recommendation systems (Wu et al., 2024; Xue et al., 2023). However, a fundamental challenge remains: the preference alignment of a pretrained model is inherently tied to its training data. This often leads to a mismatch when the model is applied to downstream tasks that require personalized or diverse preferences (Jang et al., 2023; Zhang et al., 2025b;d). While this gap can be bridged through task-specific post-training (Zhang et al., 2025b; Li et al., 2025b), such methods demand significant investment in data curation and computational resources.

To circumvent the costs of post-training, we explore aligning models at inference time. While some approaches modify the policy distribution at the token level to reflect user preferences (Zhang et al., 2025c; Gao et al., 2024), they tend to be computationally intensive and scale poorly. A more promising direction is Test-Time Scaling (TTS) (OpenAI, 2024; Muennighoff et al., 2025; Beeching et al., 2025), where models leverage additional computation during generation to enhance output quality. However, existing TTS research has predominantly focused on domains such as mathematics and coding, where the correctness can be easily verified (OpenAI, 2024). Applying TTS to preference alignment is more challenging, as the quality of a response is holistic and not reducible to a simple verifiable answer. This raises a critical question: how can we effectively guide a TTS framework to evaluate and improve responses for complex preference alignment tasks?

In this work, we address this challenge by framing the TTS process as a realignment problem. We posit that while a pretrained model possesses general instruction-following abilities, its original training objective may not be optimal for a specific user's needs. An inference-time realignment process can rescale the importance of user preference to generate a more aligned response. As illustrated in Figure 1, when a user asks for enjoyable ways to study math but expresses a dislike

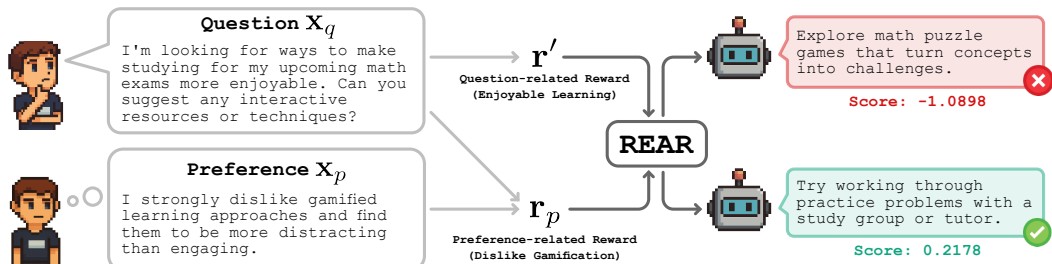

Figure 1: A motivating example of REAR. The method realigns its response from a gamified suggestion to a collaborative one when selecting candidate responses according to REAR scores.

for gamification, a TTS method might generate multiple responses. Some responses may only focus on answering the question of "enjoyable learning approaches", while the preferred responses should also align with the preference on "dislike gamification". Our REAlignment Reward (REAR) is designed to capture the preference alignment capabilities. Specifically, we decompose the reward of a pretrained LLM into a question-related component and a preference-related component. REAR then rescales the preference component to acquire a realigned reward value, allowing us to score the candidate responses and thus effectively select the most aligned option. We further show that REAR can be efficiently computed as a linear combination of policy probabilities, and then incorporate REAR into two TTS methods: a simple best-of-N sampling strategy (Stiennon et al., 2020) and a more sophisticated tree-search algorithm DVTS (Beeching et al., 2025). The contributions of this paper are summarized as follows:

- We formalize test-time preference alignment as a realignment problem and propose REAR, a computationally efficient reward from a decomposed preference alignment objective.

- We develop two scalable TTS methods guided by REAR: a best-of-N sampling approach and a DVTS-based search algorithm.

- Extensive experiments on preference alignment and role-play benchmarks show that our REAR-guided TTS methods outperform existing test-time alignment approaches.

## 2 PRELIMINARIES

In this section, we first formalize the text generation problem as a Markov Decision Process (MDP) (Puterman, 1994; Sutton & Barto, 2018) at the token level. The MDP model enables us to see how we can apply modern reinforcement learning (RL) algorithms (Schulman et al., 2017; 2015) to text generation problems. Then we will provide a view of reinforcement learning from human feedback (RLHF) (Ouyang et al., 2022) from the perspective of rewards in the given MDP model.

### 2.1 TOKEN-LEVEL MDP FOR TEXT GENERATION

We can model the text generation process as an MDP according to Ramamurthy et al. (2023). The MDP can be defined as a tuple $\mathcal{M} = \langle \mathcal{S}, \mathcal{A}, \mathcal{P}, r, \gamma, \rho, T \rangle$, where $\mathcal{S}$ is the state space and $\mathcal{A}$ is the action space defined as the vocabulary of a language model, where each action is a token in the vocabulary. We use $\pi(a \mid s)$ to denote the policy, i.e., an LLM, that provides a distribution of actions given the state $s$. At the beginning of text generation, the prompt $x = (x_1, x_2, \cdots, x_m)$ of length $m$ is sampled from the initial distribution $\rho(s)$ as the initial state $s_0$, while we use the policy $\pi(\cdot \mid s_t)$ to sample an action $a_t$ at each time step $t \in \{1, \ldots, T\}$. The MDP thus transits to the next state $s_{t+1} \sim \mathcal{P}(\cdot \mid s_t, a_t)$ according to the transition function $\mathcal{P}$. The transition function is a deterministic function satisfying $\mathcal{P}(s_{t+1} \mid s_t, a_t) = 1$ when $s_{t+1} = s_t \oplus a_t$, where $\oplus$ is the concatenation operation. The reward function $r(s_t, a_t)$ is given at each time step $t$, where the model maximizes the discounted cumulative reward with a discount factor $\gamma$. The episode terminates when the model generates an end-of-sequence token defined in the vocabulary or exceeds the maximum length $T$. We assume that the early-stop sequence is also padded to length $T$ for notational simplicity.

## 2.2 REINFORCEMENT LEARNING FROM HUMAN FEEDBACK

The main objective of reinforcement learning from human feedback (RLHF) is to find a policy that can maximize the expected cumulative reward of the defined MDP. Classical RLHF methods (Ouyang et al., 2022; Bai et al., 2022) usually consider learning a reward model to turn human preferences into reward signals. The objective can be formulated as follows:

$$\max_{\pi} \mathbb{E}_{s_0 \sim \rho, a_t \sim \pi(\cdot|s_t), s_{t+1} \sim \mathcal{P}(s_t, a_t)} \left[ \sum_{t=0}^{T} \gamma^t (r(s_t, a_t) - \beta D_{\mathrm{KL}}(\pi(\cdot|s_t) \| \pi_{\mathrm{ref}}(\cdot|s_t))) \right], \quad (1)$$

where we use $D_{\mathrm{KL}}$ to denote the Kullback-Leibler (KL) divergence and $\beta$ is a hyper-parameter to limit the divergence between the policy to be learned $\pi$ and a reference policy $\pi_{\mathrm{ref}}$. The reference policy usually comes from the base model that is used to initialize RL training. Following Li et al. (2025b), we can convert this objective from the perspective of maximum entropy RL (Haarnoja et al., 2018) according to the following proposition.

**Proposition 2.1.** *The optimization problem in Equation* (1) *is equivalent to*

$$\max_{\pi} \mathbb{E}_{s_0 \sim \rho} \left[ \mathbb{E}_{a_0 \sim \pi(\cdot|s_0)} [Q^{\pi}(s_0, a_0) + \beta \mathcal{H}(\pi(\cdot|s_0))] \right], \quad (2)$$

*where $\mathcal{H}(\pi(\cdot|s_t)) = \mathbb{E}_{a_t \sim \pi(\cdot|s_t)}[-\log \pi(a_t|s_t)]$ is the entropy of $\pi$ in the state $s_t$, and*

$$Q^{\pi}(s_0, a_0) = \mathbb{E}_{s_t \sim \mathcal{P}(s_0, a_0), a_t \sim \pi(\cdot|s_t)} \left[ r'(s_0, a_0) + \sum_{t=1}^{T} \gamma^t (r'(s_t, a_t) + \beta \mathcal{H}(\pi(\cdot|s_t))) \right]. \quad (3)$$

*is the soft-Q function of the policy $\pi$. The reshaped reward $r'(s, a) = r(s, a) + \beta \log \pi_{\mathrm{ref}}(a|s)$. The soft-Q function $Q^{\pi}$ satisfies the following Bellman equation:*

$$Q^{\pi}(s, a) = r'(s, a) + \gamma \mathbb{E}_{s' \sim \mathcal{P}(s, a), a' \sim \pi(\cdot|s')} [Q^{\pi}(s', a') + \beta \mathcal{H}(\pi(\cdot|s'))] = r'(s, a) + \gamma V^{\pi}(s'). \quad (4)$$

*We denote $V^{\pi}(s) = \mathbb{E}_{a \sim \pi(\cdot|s)}[Q^{\pi}(s, a) + \beta \mathcal{H}(\pi(\cdot|s))]$ as the value function of policy $\pi$.*

We defer the proof to Appendix A.1. Here Proposition 2.1 shows that the RLHF objective can be converted to the maximum entropy RL problem under the reward $r'$. As this optimization manner is widely used in LLM research, we can thus use various open-source LLMs to address our preference realignment problem described in the following section.

## 3 TEST-TIME REALIGNMENT THROUGH REWARD DECOMPOSITION

In this section, we detail our method for test-time preference realignment. We begin by introducing the theoretical foundation of our approach: a reward decomposition that separates the model's objective into question-related and preference-related components. Based on this, we derive our REAlignment Reward (REAR), a score that allows us to control the emphasis on user preference. Finally, we show how REAR can be integrated into standard test-time scaling (TTS) algorithms like best-of-N sampling (Stiennon et al., 2020) and DVTS (Beeching et al., 2025) to produce more aligned responses.

### 3.1 REALIGNMENT REWARD (REAR)

In a preference alignment task, an LLM receives a question prompt $x_q$ and a preference prompt $x_p$. The concatenated prompt $x = x_q \oplus x_p$ is used to generate a response. To formalize this into a token-level MDP form, we define the state $s$ as the sequence that contains the full prompt $x$ and the generated answer, and the state $s^q$ as the sequence that contains only the question and the generated answer. Therefore, we can obtain two reward terms $r'(s, a)$ and $r'(s^q, a)$, which represent the reward when considering the full prompt and only the question part, respectively. Although we cannot directly access these rewards, there exists a relationship between these two terms. Intuitively, the reward $r'(s, a)$ should contain both the reward $r'(s^q, a)$ which only considers the question, and an additional reward that focuses on the preference part, which forms the following equation.

$$r'(s, a) = r'(s^q, a) + \alpha r_p(s, a), \quad (5)$$

where $r_p(s, a)$ is a preference-related reward that reflects how the chosen action aligns with the given preference. Here we introduce a linear combination to decompose the reward $r'(s, a)$ into the question-related reward $r'(s^q, a)$ and the preference-related reward $r_p(s, a)$.

**Lemma 3.1.** *The policy $\pi(a|s)$ when taking the full prompt $x$ as input is the optimal policy of the following optimization problem under the decomposition in Equation* (5)*:*

$$\max_{\hat{\pi}} \mathbb{E}_{s_0 \sim \rho, a_t \sim \hat{\pi}(\cdot|s_0), s_{t+1} \sim \mathcal{P}(s_t, a_t)} \left[ \sum_{t=0}^{T} \gamma^t \left( r_p(s_t, a_t) - \frac{\beta}{\alpha} D_{\text{KL}}(\hat{\pi}(\cdot|s_t) \| \pi(\cdot|s_t^q)) \right) \right], \quad (6)$$

*where $s_t^q$ is the corresponding question-only state of $s_t$.*

The proof can be found in Appendix A.2. Lemma 3.1 reveals that the original policy $\pi(a|s)$ implicitly maximizes the preference-related reward $r_p(s, a)$ subject to a constraint on the KL-divergence from the distribution of the question-only policy. This framing of Reward Decomposition is essential. Unlike heuristic strategies such as simple policy interpolation, proving that the base model inherently optimizes a specific reward structure allows us to treat the derived REAR score as a valid value function. This theoretical foundation validates the use of lookahead search algorithms like DVTS, which require a consistent reward signal, rather than being limited to simple sampling heuristics. This framing suggests a clear path to realignment: if we could control this trade-off at test time, we could steer the generation to be more or less aligned with the preference. To this end, we introduce a new, flexible coefficient $\hat{\alpha}$ to re-weight the preference component, defining our realignment reward as:

$$r_{\text{REAR}}(s, a) = r'(s^q, a) + \hat{\alpha} r_p(s, a). \quad (7)$$

By adjusting $\hat{\alpha}$ at test time, we can modulate the influence of the preference reward, steering the generation towards responses that are more aligned with a user's specific needs, without altering the underlying model. The challenge here is that $r_{\text{REAR}}(s, a)$ is defined in terms of unobserved reward components. Fortunately, the framework of maximum entropy RL (Haarnoja et al., 2018; Li et al., 2025a) allows us to express this reward in a computable form based on policy probabilities.

**Lemma 3.2.** *The realignment reward $r_{\text{REAR}}(s, a)$ keeps policy-optimality with the following proxy reward:*

$$\hat{r}_{\text{REAR}}(s, a) = \frac{(\alpha - \hat{\alpha})\beta}{\alpha} \log \pi(a \mid s^q) + \frac{\hat{\alpha}\beta}{\alpha} \log \pi(a \mid s). \quad (8)$$

Intuitively, this substitution is grounded in Maximum Entropy RL, where the optimal policy follows a Boltzmann distribution $\pi^*(a|s) \propto \exp(Q^*(s, a)/\beta)$. Since the Q-function represents the long-term cumulative reward, the log-probability of the policy is directly proportional to the reward plus value function terms. This allows us to mathematically recover the implicit reward optimizing the policy from the log-probabilities themselves, providing a dense, token-level signal without training a separate reward model. We defer the detailed proof to Appendix A.3, which shows that the difference between the two rewards is a potential-based shaping term (Ng et al., 1999).

## 3.2 Test-time Scaling with REAR

Our goal is to find a policy that maximizes the expected discounted REAR at inference time. According to Lemma 3.2, this is equivalent to maximizing the expected discounted proxy reward $\hat{r}_{\text{REAR}}(s, a)$. Since the optimal policy is invariant to positive scaling of the reward function, we can simplify $\hat{r}_{\text{REAR}}(s, a)$ by omitting the constant factor $\beta$ to derive the following score function:

$$S(s, a) = (1 - \lambda) \log \pi(a \mid s^q) + \lambda \log \pi(a \mid s), \quad (9)$$

where we set $\lambda = \frac{\hat{\alpha}}{\alpha} > 0$ as a hyper-parameter. This concise formulation reveals how we integrate the LLM preferences that are hard to verify by encoding its output probability to a token-level reward. Intuitively, $\lambda > 1$ indicates that the preference is more important in the real case than when the model is trained and $\lambda < 1$ will reduce the importance of the preference. When $\lambda = 1$, the result is equivalent to directly using the original LLM for inference. In our experiments, we find that choosing a relatively large $\lambda$ will yield better performance on benchmark scores in most tasks.

This score can be extended to a response trajectory $\tau = (s_0, a_0, \ldots, s_T, a_T)$ across multiple tokens in the form of a cumulative score:

$$S(\tau) = \sum_{t=0}^{T} \gamma^t S(s_t, a_t). \quad (10)$$

Since $\tau$ can represent either a complete or partial response, this formulation allows for flexible integration with various TTS methods. We explore two such methods:

**Best-of-N (BoN) with REAR.** We simply sample $N$ responses and calculate the REAR score for each response. Then we select the response with the highest score as the final response.

**Diverse Verifier Tree Search (DVTS) with REAR.** We use the DVTS (Beeching et al., 2025) algorithm to select a final response, where the response generated step-by-step in a tree search manner and selected according to the REAR score.

Compared to external or generative reward models (Lambert et al., 2025; Liu et al., 2024a; Zhang et al., 2024; Mahan et al., 2024), REAR solves the preference alignment problem by solely rescaling its inherent preferences, without requiring extra training, external model calls or extra generation steps. This makes REAR highly flexible and readily deployable in a plug-and-play manner across almost any LLM. Moreover, since REAR provides a token-level reward formulation, it can be applied to partial responses, enabling its use with advanced TTS algorithms like DVTS, which is not valid for general reward models that can only perform effective evaluations with the whole response.

## 4 RELATED WORK

**Preference Alignment** Aligning LLMs with human preferences is a central challenge in AI safety and usability. Early and prominent approaches rely on training-based methods, particularly reinforcement learning from human feedback (RLHF) (Ouyang et al., 2022; Bai et al., 2022), where a reward model is trained on human preference data to fine-tune a base model. Subsequent work has sought to simplify this pipeline (Rafailov et al., 2023) bypassing the need for an explicit reward model. Other approaches focus on creating specialized data curricula (Zhang et al., 2025b) or maintaining original capabilities when adapting to new preferences (Li et al., 2023; Wang et al., 2025; Li et al., 2025b). While effective, these training-based methods often require extensive data and are computationally expensive. This motivates a shift towards test-time alignment methods that adapt model behavior without updating weights. For instance, Zhang et al. (2025c) and Gao et al. (2024) propose techniques to modify the model's output distribution at each generation step to better align with given preferences. Our work builds on this line of research but focuses on scaling the alignment process through a novel reward formulation within a TTS framework rather than direct policy modification, which provides a stable and scalable performance improvement.

**Test-time Scaling** Test-time scaling (TTS) aims to improve model performance by allocating more computational resources during inference, realized by extended thinking (OpenAI, 2024; Guo et al., 2025; Muennighoff et al., 2025) or parallel searching (Wang et al., 2024a; Comanici et al., 2025; Huang & Yang, 2025). This paradigm has been particularly successful in domains where answers can be easily extracted and verified, such as mathematical and coding problems (OpenAI, 2024; Zhang et al., 2025a), where researchers adopt self-consistency (Wang et al., 2023; Li et al., 2024) and use explicit verifiers such as process-based reward models (Lightman et al., 2024; Wang et al., 2024b) with sophisticated search algorithms (Wei et al., 2022; Yao et al., 2023; Wang et al., 2024a) that explore different reasoning paths. However, applying TTS to open-ended preference alignment tasks is challenging due to the absence of a simple, verifiable ground truth. Generative reward models (Zhang et al., 2024; Mahan et al., 2024; Liu et al., 2025) are proposed for their ability to verify an answer through the generation process but still face challenges on computational efficiency and accuracy. Recent study (Li et al., 2025a) indicates that the LLM itself is an implicit reward model, supporting the validity of policy probabilities as rewards. Our approach differs by deriving a specialized reward, REAR, that is specifically designed for preference realignment and can be integrated into various TTS algorithms, bridging the gap between TTS for verifiable reasoning and TTS for subjective preference alignment. Unlike methods like ARGS (Khanov et al., 2024) or IVG (Liu et al., 2024b) which rely on training and hosting external Reward Models or value heads, REAR is fully training-free and derives its signal solely from the base model's internal probabilities. This allows REAR to extend TTS to open-ended domains where no ground-truth verifiers exist.

## 5 EXPERIMENTS

In this section, we investigate the efficacy of REAR-guided test-time sampling (TTS) on existing preference alignment tasks. We first describe our experimental setup in Section 5.1. Then in Section 5.2, we specifically seek to determine whether our proposed hyperparameter, $\lambda$, can effectively

control the degree of alignment with user preferences. In Section 5.3, we evaluate the performance of our method against several baselines, including other test-time preference alignment methods and TTS approaches that use different reward forms. In Section 5.4, we further show the scaling performance of our methods and analyze the robustness and efficiency of our method.

## 5.1 EXPERIMENTAL SETUP

**Evaluation Benchmarks**   To evaluate the preference alignment capabilities of different methods, we use three recent benchmarks:

- **PrefEval** (Zhao et al., 2025) requires the LLM to generate personalized responses across conversations according to the user's previously stated preferences, which provides a comprehensive evaluation of the LLM's capability on inferring, remembering, and applying the user preference to multi-turn conversations. The PrefEval benchmark contains three data types, including explicit preference, implicit choice, and implicit preference.
- **Multifaceted Bench** (Lee et al., 2024) is designed to evaluate whether the LLM can generate context-specific responses tailored to user preferences. Each sample is paired with synthetic system messages and reference answers.
- **PingPong** (Gusev, 2024) evaluates the role-playing capabilities of LLMs through a multiturn conversation. As role-playing can be framed as a preference alignment problem, we use this benchmark to assess our method's effectiveness in this practical scenario.

**Baselines**   Beyond greedy decoding, several methods can align model outputs with human preferences. We compare REAR against baselines from two main categories, with implementation details provided in Appendix C:

- **Test-time preference alignment methods**.   We include two representative methods: Amulet (Zhang et al., 2025c) and Linear Alignment (LA) (Gao et al., 2024). These methods align generations with preferences by modifying the token-level generation probability distribution.
- **Test-time Sampling with Other Rewards**. Like our method, these baselines use best-of-N (BoN) sampling but employ different reward sources. We consider two variants: one using an external reward model (External RM) and another using the generative model itself as a reward source (GenRM). For the external RM, we use the Skywork-Reward-Llama-8B model (Liu et al., 2024a) due to its strong performance on RewardBench (Lambert et al., 2025) and its comparable size to our base model.

For our main experiments, we use `Qwen2.5-7B-Instruct` (Yang et al., 2024) as the base model. We employ the SGLang inference engine (Zheng et al., 2024) for response generation, maintaining consistent sampling parameters across all methods except for Amulet and Linear Alignment, for which we use the authors' original implementation (Zhang et al., 2025c). We use $N = 16$ samples for BoN methods in our experiments or equivalent sampling size for DVTS. Further implementation details are provided in Appendix C.

## 5.2 CONTROLLABLE REALIGNMENT WITH $\lambda$

As established in our methodology, the hyperparameter $\lambda$ governs the strength of preference alignment by scaling the preference-related reward. A larger $\lambda$ directs more attention to this reward while a smaller $\lambda$ may not sufficiently align the model with user preferences but focuses more on answering the question. In this section, we investigate the impact of $\lambda$ on benchmark performance. We conduct experiments on the PrefEval benchmark, evaluating both Best-of-N and DVTS with REAR across a range of $\lambda$ values.

We focus on two data types from PrefEval: explicit preference and implicit choice. Although derived from the same source data, they employ different prompting and evaluation protocols. For the explicit preference task, the model must generate a response that adheres to a given system preference prompt. An external LLM judge evaluates the response quality based on multiple rubrics, including helpfulness, preference violation, consistency, and hallucination. We report the average score across

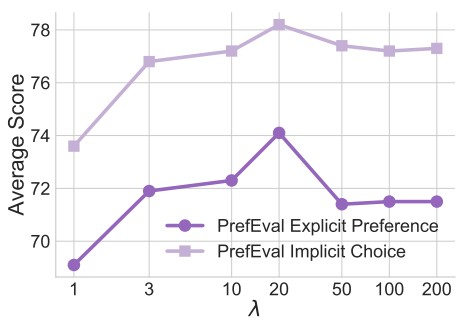
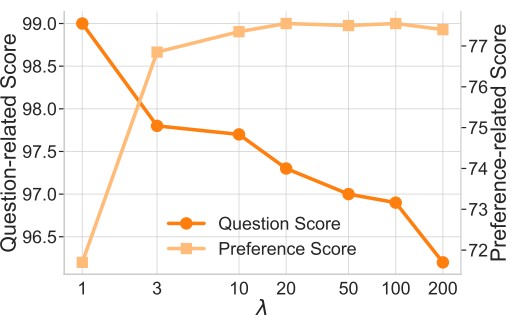

Figure 2: Benchmark scores of REAR-guided TTS methods on PrefEval explicit preference and implicit choice data with different $\lambda$ values.

Figure 3: Scores on questions and preference of REAR-guided TTS methods on PrefEval explicit preference data with different $\lambda$ values.

Table 1: Performance comparison of REAR-guided TTS methods and other baselines on various preference alignment benchmarks. Bold values indicate the best performance on the corresponding benchmark.

| Benchmark | DVTS w/ REAR (Ours) | BoN w/ REAR (Ours) | Greedy | External RM | GenRM | Amulet | LA |
|---|---|---|---|---|---|---|---|
| *PrefEval Scores* | | | | | | | |
| Explicit Preference | **77.7** | 74.1 | 67.0 | 73.4 | 69.0 | 68.5 | 64.2 |
| Implicit Choice | **78.6** | 78.2 | 71.5 | 78.3 | 74.7 | 70.4 | 78.0 |
| Implicit Preference | **19.1** | 16.2 | 12.0 | 17.0 | 12.9 | 13.1 | 12.8 |
| Multifaceted Bench | **76.8** | 76.3 | 75.3 | 76.5 | 76.1 | 75.4 | 75.6 |
| *Ping-Pong Bench* | | | | | | | |
| Score | 3.03 | **3.07** | 2.97 | 2.97 | 3.01 | 2.87 | 3.01 |
| Stay in Character Score | 2.19 | **2.35** | 2.01 | 2.10 | 2.09 | 2.07 | 2.13 |
| Fluency Score | 4.67 | 4.50 | **4.88** | 4.52 | 4.76 | 4.47 | 4.70 |
| Entertaining Score | 2.24 | **2.36** | 2.02 | 2.27 | 2.18 | 2.09 | 2.20 |

all rubrics. In contrast, the implicit choice task presents preferences within a multi-turn conversation, from which the model must infer the user's inclination. The evaluation is a multiple-choice question where the model selects the most preferred response out of four options, and performance is measured by accuracy.

**Benchmark Scores with Different $\lambda$** As shown in Figure 2, the performance of BoN with REAR on both PrefEval explicit preference and implicit choice data varies with $\lambda$. The scores for both data types follow a similar trend: they first increase and then decrease as $\lambda$ grows. Optimal performance on both tasks is achieved consistently at $\lambda = 20.0$, with lower scores observed for both smaller and larger values of $\lambda$. We also find similar trends when adjusting $\lambda$ in other tasks and the results are deferred to Appendix E.

**Analysis on Generated Responses** To understand this non-monotonic relationship, we analyze how $\lambda$ affects different aspects of response quality. The detailed rubrics from the PrefEval explicit preference task allow us to disentangle performance into two components: general response quality and preference alignment. We use the "helpfulness" score to measure the former and the average of "preference violation" and "preference acknowledgement" scores for the latter. As illustrated in Figure 3, these two components exhibit monotonic trends with respect to $\lambda$. As $\lambda$ increases, the preference-related score improves, while the question-related score (helpfulness) declines. This trade-off explains why simply increasing $\lambda$ does not guarantee better overall performance; an excessively large $\lambda$ compromises the model's fundamental ability to provide helpful answers, thereby reducing the overall quality of the response.

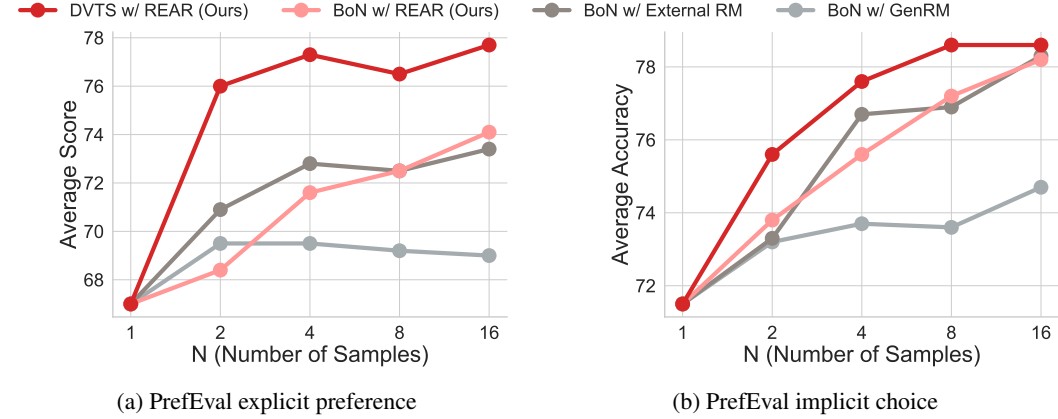

(a) PrefEval explicit preference

(b) PrefEval implicit choice

Figure 4: Scaling performance on the PrefEval benchmark with varying numbers of samples ($N$) for different methods. We use the average LLM-evaluated scores for the explicit preference task (left) and the accuracy of selected choices for the implicit choice task (right).

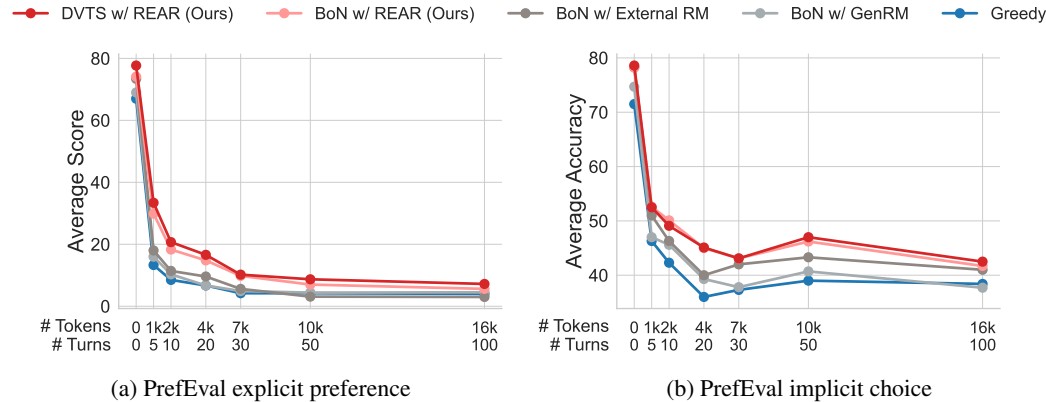

(a) PrefEval explicit preference

(b) PrefEval implicit choice

Figure 5: Long-context performance of REAR-guided TTS methods and other baselines on the explicit preference data and implicit choice data from the PrefEval benchmark with augmented conversation turns. The x-axis indicates the number of conversation turns and the estimated total number of tokens for the augmented conversational data. We use the average LLM-evaluated scores for the explicit preference task (left) and the accuracy of choices for the implicit choice task (right).

## 5.3 PERFORMANCE COMPARISONS

We compare our methods against the baselines on the PrefEval, Multifaceted, and Ping-Pong benchmarks. As shown in Table 1, both BoN with REAR and DVTS with REAR outperform all baselines on most benchmarks, demonstrating strong performance on both accuracy-based (PrefEval implicit choice) and LLM-evaluated tasks. The BoN baseline using an external RM also performs competitively, likely because the external model provides a valuable additional reward signal to select the best response. In contrast, using the generative model itself as a reward model (GenRM) does not yield significant improvement, suggesting that the model struggles to reliably verify its own responses. In addition, the test-time preference alignment methods, including Amulet and LA, also underperform on these benchmarks.

On a benchmark-specific level, we observe a significant performance drop on the PrefEval implicit preference task compared to the other two PrefEval tasks, which is consistent with previous findings (Zhao et al., 2025). Interestingly, on the Ping-Pong benchmark, all TTS methods achieve higher scores on the "stay-in-character" and "entertaining" rubrics, but decrease the "fluency" score, where greedy decoding performs best. This suggests that TTS methods prioritize role-playing traits at the

expense of fluency. For the multifaceted benchmark, while we do not report detailed rubric scores since they differ from specific samples, our DVTS variant again outperforms the baselines.

## 5.4 ROBUSTNESS AND EFFICIENCY OF REAR-GUIDED TTS

**Scaling Performance**   We investigate how the performance of our method scales with the number of samples ($N$). As shown in Figure 4, performance on the PrefEval explicit preference and implicit choice datasets improves as $N$ increases, with diminishing returns for larger values. Our BoN approach with REAR demonstrates scaling performance comparable to the variant using an external RM. The DVTS variant achieves stronger performance with a smaller sampling budget, highlighting the efficiency of its step-by-step tree search approach.

**Robustness on Long-context Input**   A key advantage of REAR is that it is derived directly from the generation process, making it inherently robust. This becomes particularly evident with out-of-distribution inputs, such as long-context prompts. Following the methodology from Zhao et al. (2025), we evaluate robustness by augmenting conversations with additional turns inserted between the preference context and the question. As shown in Figure 5, our methods consistently outperform the baselines across various context lengths. Test-time alignment baselines including Amulet and LA are excluded due to out-of-memory errors on long-context data. The performance of BoN with an external RM and GenRM degrades significantly

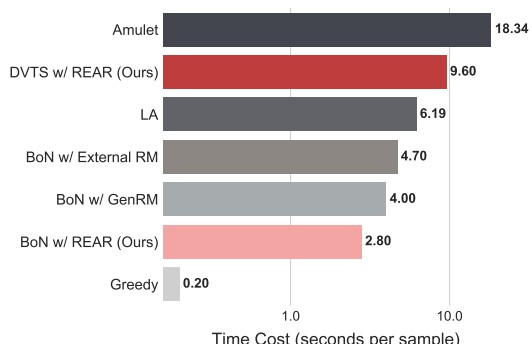

Figure 6: Time cost of different methods on the PrefEval explicit preference task.

on long-context inputs, occasionally falling below the greedy baseline, since the augmented data lies outside the external RM's training distribution, leading to unreliable reward signals.

**Efficiency of REAR**   REAR offers significant efficiency gains over baselines that rely on external reward models. We report the inference cost of REAR-guided methods compared to other baselines on the PrefEval explicit preference task in Figure 6, using a node of 8 NVIDIA GPUs with 96GB memory, by calculating rewards from the model's internal probabilities, REAR avoids the substantial computational overhead of loading and executing additional models. This makes REAR-guided methods not only more efficient but also easier for deployment.

## 6 CONCLUSION

In this work, we introduced the REAlignment Reward (REAR), a novel and efficient reward that realigns LLM to user preferences at test time. By decomposing the underlying reward into question-related and preference-related portions, we can calculate REAR directly from the model's own policy probabilities. We further integrate two test-time scaling methods, best-of-N sampling and DVTS, into REAR, enabling controllable and effective preference realignment without any model training. Extensive experiments show that REAR-guided TTS methods significantly outperforms both existing test-time alignment techniques and TTS methods guided by other rewards across a range of preference alignment benchmarks. Our work provides a controllable and scalable solution for personalizing LLM interactions and enables test-time scaling to more subjective, open-ended domains without the need of other models.

Despite these promising results, our work has several limitations. First, the performance of REAR is dependent on the hyperparameter $\lambda$, which may differ from data samples. Although we show that the optimal range of $\lambda$ is relatively consistent, pre-evaluation on a validation dataset or selecting appropriate values with heuristic methods can help further improve the performance. Second, while TTS is more lightweight than fine-tuning a model, it still introduces significant computational overhead at inference time. Identifying the sweet spot of REAR-guided TTS methods without incurring excessive computational cost can be a promising direction.

## ETHICS STATEMENT

The authors of this paper have adhered to the ICLR Code of Ethics. Our work focuses on improving the alignment of large language models with user preferences, which we believe is a crucial step toward developing safer and more helpful AI systems. We acknowledge that, like any alignment technique, our method could potentially be misused to align models with harmful or unethical preferences. However, the core principles of our approach are designed to provide controllable and transparent realignment, which can also serve as a tool for safety researchers to better understand and mitigate undesirable model behaviors. The experiments are conducted on publicly available benchmarks, which do not contain personally identifiable information. We encourage responsible use of this technology and further research into robust safety guardrails for preference alignment techniques. Our use of large language models for evaluation was conducted via standard APIs, and we acknowledge the associated computational and environmental costs.

## REPRODUCIBILITY STATEMENT

The code and data are provided in the supplementary material, while our used model is publicly available. The README file within the code submission contains detailed instructions on setting up the environment and running experiments presented in the paper. Appendix C also provides a comprehensive description of the implementation details, including the base model used, key hyperparameters, and the setup for all baseline methods. Appendix D details the evaluation protocols for each benchmark.

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

## A DEFERRED PROOFS

### A.1 PROOF OF PROPOSITION 2.1

The objective of RLHF is to maximize the expected discounted reward regularized by the KL divergence between the learned policy $\pi$ and a reference policy $\pi_{\text{ref}}$:

$$\max_{\pi} \mathbb{E}_{\tau \sim \pi} \left[ \sum_{t=0}^{T} \gamma^t (r(s_t, a_t) - \beta D_{\text{KL}}(\pi(\cdot|s_t) \| \pi_{\text{ref}}(\cdot|s_t))) \right], \tag{11}$$

where the expectation is over trajectories $\tau = (s_0, a_0, s_1, \dots)$ sampled from the policy $\pi$.

First, we expand the KL divergence term:

$$D_{\text{KL}}(\pi(\cdot|s_t) \| \pi_{\text{ref}}(\cdot|s_t)) = \mathbb{E}_{a_t \sim \pi(\cdot|s_t)}[\log \pi(a_t|s_t) - \log \pi_{\text{ref}}(a_t|s_t)]. \tag{12}$$

Substituting this into the objective and taking the expectation over actions inside the summation gives:

$$\max_{\pi} \mathbb{E}_{\tau \sim \pi} \left[ \sum_{t=0}^{T} \gamma^t (r(s_t, a_t) - \beta(\log \pi(a_t|s_t) - \log \pi_{\text{ref}}(a_t|s_t))) \right]. \tag{13}$$

We can rearrange the terms within the summation:

$$\max_{\pi} \mathbb{E}_{\tau \sim \pi} \left[ \sum_{t=0}^{T} \gamma^t ((r(s_t, a_t) + \beta \log \pi_{\text{ref}}(a_t|s_t)) - \beta \log \pi(a_t|s_t)) \right]. \tag{14}$$

Let us define a reshaped reward function $r'(s_t, a_t) = r(s_t, a_t) + \beta \log \pi_{\text{ref}}(a_t|s_t)$. Additionally, we recognize that the term $-\mathbb{E}_{a_t \sim \pi(\cdot|s_t)}[\log \pi(a_t|s_t)]$ is the entropy of the policy, denoted by $\mathcal{H}(\pi(\cdot|s_t))$. With these substitutions, the objective becomes:

$$\max_{\pi} \mathbb{E}_{\tau \sim \pi} \left[ \sum_{t=0}^{T} \gamma^t (r'(s_t, a_t) + \beta \mathcal{H}(\pi(\cdot|s_t))) \right]. \tag{15}$$

This is the standard objective for maximum entropy reinforcement learning. The expected return in this framework is the definition of the soft value function $V^\pi(s_0)$. The objective can thus be written in terms of the soft Q-function and entropy at the initial state, which is equivalent to the formulation in Equation (2).

### A.2 PROOF OF LEMMA 3.1

Let $\pi_q(\cdot|s) = \pi(\cdot|s^q)$. In maximum entropy reinforcement learning, the optimal policy $\pi^*$ is related to the soft Q-function by $\log \pi^*(a|s) = (Q^{\pi^*}(s, a) - V^{\pi^*}(s))/\beta$, where $V^{\pi^*}(s)$ is the soft value function. The Q-function satisfies the Bellman equation $Q^{\pi^*}(s, a) = r'(s, a) + \gamma V^{\pi^*}(s')$. Combining these, we can express the reshaped reward as:

$$r'(s, a) = \beta \log \pi^*(a|s) + V^{\pi^*}(s) - \gamma V^{\pi^*}(s'). \tag{16}$$

The term $V^{\pi^*}(s')$ depends on the action $a$ through the next state $s'$. For this proof, we adopt the common approximation that this value is constant with respect to $a$, which is reasonable when a single token has a limited impact on the total future reward. Under this approximation, we can apply this relation to our two policies, $\pi(a|s)$ and $\pi_q(a|s)$:

$$r'(s, a) = \beta \log \pi(a|s) + C_1(s), \tag{17}$$

$$r'(s^q, a) = \beta \log \pi_q(a|s) + C_2(s), \tag{18}$$

where $C_1(s)$ and $C_2(s)$ are terms independent of the current action $a$. Using the reward decomposition from Equation (5), $r'(s, a) = r'(s^q, a) + \alpha r_p(s, a)$, we can substitute the expressions above:

$$\beta \log \pi(a|s) + C_1(s) = \beta \log \pi_q(a|s) + C_2(s) + \alpha r_p(s, a). \tag{19}$$

Rearranging the terms, we find the optimality condition for $\pi(a|s)$:

$$\log \pi(a|s) - \log \pi_q(a|s) = \frac{\alpha}{\beta} r_p(s, a) + \text{terms independent of } a. \tag{20}$$

Now, consider the KL-regularized optimization problem from the lemma. The per-step objective to maximize at a state $s$ is:

$$\max_{\hat{\pi}} \mathbb{E}_{a \sim \hat{\pi}(\cdot|s)}[r_p(s,a) + \gamma V(s')] - \frac{\beta}{\alpha} D_{\text{KL}}(\hat{\pi}(\cdot|s) \| \pi_q(\cdot|s)), \tag{21}$$

where $V(s')$ is the value of the next state. The solution $\hat{\pi}^*$ to this optimization is well-known:

$$\hat{\pi}^*(a|s) \propto \pi_q(a|s) \exp\left(\frac{\alpha}{\beta}(r_p(s,a) + \gamma V(s'))\right). \tag{22}$$

Taking the logarithm, we find the optimality condition for $\hat{\pi}^*$:

$$\log \hat{\pi}^*(a|s) - \log \pi_q(a|s) = \frac{\alpha}{\beta} r_p(s,a) + \text{terms independent of } a. \tag{23}$$

Since the optimality conditions in Equation (20) and Equation (23) are identical, their solutions must be identical. Therefore, $\pi(a|s) = \hat{\pi}^*(a|s)$, which proves the lemma.

### A.3 PROOF OF LEMMA 3.2

We start from the definition of the realignment reward from Equation (7):
$$r_{\text{REAR}}(s,a) = r'(s^q,a) + \hat{\alpha} r_p(s,a). \tag{24}$$
From the reward decomposition in Equation (5), we can express the preference-related reward $r_p(s,a)$ as:
$$r_p(s,a) = \frac{1}{\alpha}(r'(s,a) - r'(s^q,a)). \tag{25}$$
Substituting this into the definition of $r_{\text{REAR}}(s,a)$, we get:

$$r_{\text{REAR}}(s,a) = r'(s^q,a) + \frac{\hat{\alpha}}{\alpha}(r'(s,a) - r'(s^q,a)) \tag{26}$$

$$= \left(1 - \frac{\hat{\alpha}}{\alpha}\right) r'(s^q,a) + \frac{\hat{\alpha}}{\alpha} r'(s,a). \tag{27}$$

Next, we relate the reshaped rewards $r'(s,a)$ and $r'(s^q,a)$ to their respective optimal policies, $\pi(a|s)$ and $\pi_q(a|s) = \pi(a|s^q)$. In maximum entropy RL, the reshaped reward can be expressed in terms of the optimal policy and the soft value functions:
$$r'(s,a) = \beta \log \pi(a|s) + V^\pi(s) - \gamma V^\pi(s'), \tag{28}$$
where $s' = s \oplus a$ is the next state. Applying this for both $r'(s,a)$ and $r'(s^q,a)$:
$$r'(s,a) = \beta \log \pi(a|s) + V^\pi(s) - \gamma V^\pi(s'), \tag{29}$$
$$r'(s^q,a) = \beta \log \pi_q(a|s) + V^{\pi_q}(s) - \gamma V^{\pi_q}(s'). \tag{30}$$
Substituting these into the expression for $r_{\text{REAR}}(s,a)$:

$$r_{\text{REAR}}(s,a) = \left(1 - \frac{\hat{\alpha}}{\alpha}\right)(\beta \log \pi_q(a|s) + V^{\pi_q}(s) - \gamma V^{\pi_q}(s'))$$
$$+ \frac{\hat{\alpha}}{\alpha}(\beta \log \pi(a|s) + V^\pi(s) - \gamma V^\pi(s')). \tag{31}$$

We can group the terms that depend on the action $a$ and those that depend only on the state $s$:

$$r_{\text{REAR}}(s,a) = \frac{(\alpha - \hat{\alpha})\beta}{\alpha} \log \pi(a \mid s^q) + \frac{\hat{\alpha}\beta}{\alpha} \log \pi(a \mid s) + Z(s,a), \tag{32}$$

where $Z(s,a)$ contains the value function terms:

$$Z(s,a) = \left(1 - \frac{\hat{\alpha}}{\alpha}\right)(V^{\pi_q}(s) - \gamma V^{\pi_q}(s')) + \frac{\hat{\alpha}}{\alpha}(V^\pi(s) - \gamma V^\pi(s')). \tag{33}$$

The term $Z(s,a)$ can be rewritten as $\Phi(s) - \gamma\Phi(s')$, where $\Phi(s) = \left(1 - \frac{\hat{\alpha}}{\alpha}\right)V^{\pi_q}(s) + \frac{\hat{\alpha}}{\alpha}V^\pi(s)$ is a potential function that depends only on the state $s$. According to the theory of potential-based reward shaping (Ng et al., 1999), adding a reward of the form $\gamma\Phi(s') - \Phi(s)$ to a base reward function does not change the optimal policy. The term $Z(s,a)$ is the negative of such a potential-based shaping reward. Therefore, the optimal policy for the full reward $r_{\text{REAR}}(s,a)$ is identical to the optimal policy for the proxy reward obtained by removing $Z(s,a)$. This justifies using only the action-dependent terms for our score function, which can be expressed as

$$\hat{r}_{\text{REAR}}(s,a) = \frac{(\alpha - \hat{\alpha})\beta}{\alpha} \log \pi(a \mid s^q) + \frac{\hat{\alpha}\beta}{\alpha} \log \pi(a \mid s). \tag{34}$$

## B  DECLARATION ON THE USE OF LLMS

We acknowledge the use of Large Language Models (LLMs) to assist in the preparation of this manuscript. Specifically, LLMs were utilized for the following tasks: (1) generating boilerplate code for experiment scripts, (2) assisting with the implementation of baselines and plotting scripts for visualizing results, (3) performing grammar and spelling checks to improve readability, and (4) proofreading the manuscript for clarity and correctness. All content, including the final text, figures, and scientific contributions, were curated and verified by the authors.

## C  IMPLEMENTATION DETAILS

Our experiments are conducted using a framework based on the SGLang inference engine (Zheng et al., 2024). For all methods, we employ the inference engine to serve the Qwen2.5-7B-Instruct (Yang et al., 2024) model, which ensures efficient and consistent response generation across all experiments. We choose this model because of its popularity and moderate performance on evaluated benchmarks, leaving enough improvement space for TTS methods.

**Calculation of REAR Scores**  The REAR score is calculated by obtaining token-level log-probabilities for each generated response under two distinct contexts: one with the full prompt including preference information and another with only the question part of the prompt. We use the SGLang frontend APIs to directly obtain the log-probabilities for each token in the response. The log-probabilities on the full prompt can be directly acquired within the text generation process, while the log-probabilities on the question part of the prompt are calculated with another simple forward process that takes the question part and the generated response as input, which can be lightweight and efficient. These two sets of log-probabilities are then combined as a weighted sum, controlled by the hyperparameter $\lambda$, to produce the final realignment score, as formulated in our methodology. To calculate the REAR score of a complete or partial response, we simply set the discount factor $\gamma = 1$ to take all the tokens into account with equal weights.

**TTS Methods**  We adapt our REAR scores to two TTS methods, best-of-N sampling (BoN) (Stiennon et al., 2020) and dynamic verifier tree search (DVTS) (Beeching et al., 2025). For BoN, we directly use the inference engine to generate multiple responses in separate requests, and then select the response with the highest REAR score. For DVTS, we use the line break as the delimiter of each tree-search step, where the algorithm will select the expanded branch of each node according to the REAR score. In our experiments, unless specified, we set the number of samples to 16 for all BoN methods including the baselines. For DVTS, we set an equivalent compute budget to the BoN method by setting its expansion width and initial tree nodes both to 4. According to Beeching et al. (2025), this setting is comparable to the $N = 16$ setting for BoN. All the generated responses are sampled using a temperature of 1.0 and the maximum generated length is set to 2048 tokens.

**Best-of-N with Generative RM (GenRM)**  This baseline leverages the base model as its own judge. Each generated response is appended with a template that prompts the model to evaluate whether the response is preferred. The final reward is calculated from the log-probability difference between the model generating "Yes" and "No". To be specific, we use the following chat template:

Listing 1: Generative Verification Prompting Template

```
System: [Preference in the data sample]

User: [Question]

Assistant: [Response]

User: Please act as an impartial judge and evaluate the
    quality of the assistant's response. A preferred response
    is helpful, harmless, and accurately follows instructions.
    Is this a preferred response? Answer 'Yes' or 'No' in the
    format 'Preferred: X'.
```

Table 2: Ablation study on the hyper-parameter $\lambda$ in REAR on different tasks from PrefEval and Multifaceted benchmarks.

| Method | $\lambda$ | | | |
|---|---|---|---|---|
| | 3 | 10 | 20 | 50 |
| *PrefEval Explicit Preference* | | | | |
| BoN w/ REAR | 71.9 | 72.3 | **74.1** | 71.4 |
| DVTS w/ REAR | **77.4** | 76.4 | 76.3 | 75.1 |
| *PrefEval Implicit Choice* | | | | |
| BoN w/ REAR | 76.8 | 77.2 | **78.2** | 77.4 |
| DVTS w/ REAR | 73.8 | 76.2 | **78.6** | 77.4 |
| *PrefEval Implicit Preference* | | | | |
| BoN w/ REAR | 14.6 | 15.1 | 15.4 | **16.2** |
| DVTS w/ REAR | 14.7 | 17.4 | **19.1** | 18.1 |
| *Multifaceted Bench* | | | | |
| BoN w/ REAR | 75.4 | 76.0 | **76.3** | 75.3 |
| DVTS w/ REAR | 74.5 | 75.3 | **76.8** | 75.6 |

```
Assistant: [Potential chain-of-thought reasoning process]
    Preferred: [Yes/No]
```

**Best-of-N with External RM**  This approach uses an external, dedicated reward model, Skywork-Reward-Llama-8B (Liu et al., 2024a), hosted on an independent inference endpoint. For each candidate response, the prompt and the response are sent to this external model, which returns a scalar reward score.

**Amulet and Linear Alignment (LA)**  We use the implementation of Amulet and LA provided by the Amulet paper (Zhang et al., 2025c) to run the experiments[1]. We do not change the default hyper-parameters of these baselines. For Amulet, experiments are run with an iteration number of 60 for test-time alignment.

## D  EVALUATION PROTOCOLS

In this section, we provide a detailed description of the evaluation protocols used for each benchmark in our experiments. Except for the PrefEval implicit choice task, which uses the accuracy on selected option as the metric, the other tasks typically adopt LLM-as-a-judge for evaluation. We choose the GPT-4.1 model as the judge by calling the OpenAI API.

**PrefEval**  The PrefEval benchmark (Zhao et al., 2025) is evaluated across its three distinct data types, each with a specific protocol. For **explicit preference**, the task is evaluated using an LLM-as-a-judge. For each generated response, a series of automated checks assesses different aspects of quality and preference alignment, including helpfulness, preference violation, consistency, and hallucination. The final score is an aggregated metric that reflects overall preference-following accuracy. The evaluation protocol for **implicit preference** is identical to that of explicit preference, using the same LLM-as-a-judge and the same set of automated checks. For **implicit choice**, this is a multiple-choice task where the model must select the best response from four options. The evaluation protocol extracts the model's choice from its generated output and compares it to the ground-truth correct answer. The final performance is measured by accuracy.

**Multifaceted Bench**  For the Multifaceted Bench (Lee et al., 2024), we also employ an LLM-as-a-judge for evaluation. The judge assesses the model's generated response based on a set of rubrics

---

[1]https://github.com/zowiezhang/Amulet

Table 3: Ablation study on $\lambda$ for REAR on Ping-Pong Bench.

| BoN w/ REAR | | | | |
| --- | --- | --- | --- | --- |
| $\lambda$ | 3 | 10 | 20 | 50 |
| Score | 2.92 | 2.97 | 3.02 | **3.07** |
| DVTS w/ REAR | | | | |
| $\lambda$ | 0.3 | 0.5 | 1 | 1.5 |
| Score | 2.88 | 2.99 | **3.03** | 2.87 |

that are provided within each data sample. It assigns a score from 1 to 5 for each rubric. The final reported score is the average of these scores across all rubrics.

**Ping-Pong Bench**    The Ping-Pong benchmark (Gusev, 2024) for role-playing is evaluated using an LLM-as-a-judge. The judge evaluates the entire conversation based on three main criteria: **stay-in-character score** (how well the model maintains its assigned persona), **entertaining score** (how engaging and entertaining the conversation is), and **fluency score** (the quality and naturalness of the language used). Each criterion is scored on a scale, and the final metric is the overall average score across these dimensions. We adopt the English version of the Ping-Pong-v2 dataset for evaluation. Differing from the original benchmark that uses gpt-4o-mini as the interrogator model to generate multi-turn data from the user side, we use the same model as our base model, i.e., Qwen2.5-7B-Instruct, as the interrogator model to avoid heavy expenses on calling the API. We note that this setting will result in slight performance degradation compared to the original benchmark. However, it is still able to capture the role-playing capabilities of the model and the comparisons are fair and valid for all evaluated methods.

# E    ADDITIONAL EXPERIMENTS ON HYPER-PARAMETER TUNING

We conduct an ablation study on the hyper-parameter $\lambda$ in REAR, which controls the weight of the value function. The results are shown in Table 2 and Table 3. The results largely confirm the observations made in Section 5.2. Across most tasks on the PrefEval and Multifaceted benchmarks, we observe a non-monotonic relationship between $\lambda$ and performance. For the majority of these tasks, the optimal performance is achieved when $\lambda$ is around 20 for both BoN and DVTS. This reinforces the idea that there is a trade-off between adhering to user preference and maintaining the general quality of the response, as an excessively high $\lambda$ can degrade helpfulness.

However, we also note some task-specific variations. For instance, on the PrefEval Explicit Preference task, DVTS achieves its best performance with a smaller $\lambda$ of 3. On the PrefEval Implicit Preference and the Ping-Pong Bench tasks, BoN with REAR shows a trend of continuously improving performance as $\lambda$ increases up to 50. This suggests that for certain tasks, particularly those requiring strong adherence to a persona (Ping-Pong) or subtle preference cues, a stronger emphasis on the preference-related reward component is beneficial.

Furthermore, the optimal range for $\lambda$ appears to depend on the specific TTS algorithm. For example, the DVTS algorithm adopts a step-by-step tree search strategy, which can be more sensitive to the preference reward. Exaggerating the preference reward may lead to suboptimal performance. In contrast, BoN methods only rate the final response after finishing generation, where a large $\lambda$ value is often preferred for the benchmark. For the Ping-Pong benchmark, DVTS achieves its peak performance at $\lambda = 1.0$, while BoN performs best with a much larger $\lambda$. This highlights that the interaction between the search strategy and the reward scaling is an important factor. In summary, while a $\lambda$ of 20 serves as a robust default for many scenarios, fine-tuning this hyper-parameter for the specific task and TTS method can unlock further performance gains.

