# OpenReview forum: "REAR: Scalable Test-time Preference Realignment through Reward Decomposition"
_ICLR.cc/2026/Conference — Submitted to ICLR 2026_

### Official Review · Reviewer_NCzj · 2025-10-20

**Soundness:** 2
**Presentation:** 3
**Contribution:** 2
**Rating:** 4
**Confidence:** 3

**Summary:**

This paper introduces REAR, a framework for aligning with user preferences at test time, specifically targeting subjective and open-ended tasks. REAR decomposes the reward function into question-related and preference-related components, enabling dynamic re-weighting of user preference without retraining. The authors show that REAR can be efficiently formulated as a linear combination of policy probabilities, making it tractable and compatible with TTS approaches such as best-of-N (BoN) sampling and Diverse Verifier Tree Search (DVTS).

**Strengths:**

1. REAR is plug-and-play, and requires no external models or retraining, making it attractive for deployment.
2. REAR outperforms both token-level preference-alignment baselines (e.g., Amulet, Linear Alignment) and TTS methods.
3. The method is grounded in a reinforcement learning formulation and provides detailed proofs.

**Weaknesses:**

1. Performance of REAR depends on the choice of the λ parameter, which may require  tuning for different questions or preferences.
2. If user preferences are not expressed in words (for example, are only implicit, behavioral, or external), the REAR method as proposed would not function without modification.
3. Heavy reliance on LLM-based evaluation for some tasks raises concerns about evaluation robustness and objectivity.

**Questions:**

1. Hyperparameter selection of λ: How robust is the method to the choice of λ across different tasks and datasets? Is there a principled or automated way to select λ at inference time, possibly in the absence of validation data?
2. It it insufficient to use only the “helpfulness” score to measure the general response quality in the Section "Analysis on Generated Responses", and also "helpfulness" can be one of the preferences to consider.

---

> ### Author Response · Authors · 2025-11-25
>
> > How robust is the method to the choice of $\lambda$? Is there a principled way to select $\lambda$?
>
> As shown in **Figure 2** and **Table 2** from Appendix E, our method is empirically quite robust to the choice of $\lambda$:
>
> - While performance follows a rise-and-fall trend, a specific choice of $\lambda = 20$ consistently yields near-optimal results across **distinct benchmarks** (PrefEval Explicit, PrefEval Implicit Choice, and Multifaceted Bench).
> - We note the optimal choices can be different for specific scenarios, e.g., the multi-turn case in the Ping-Pong Bench or short generation for DVTS. However, the change on optimal $\lambda$ is often trackable, as long responses benefit more from larger $\lambda$ and short responses prefer smaller $\lambda$.
>
> To further show the robustness of the choices, we conduct more experiments in the PrefEval implicit choice task with five runs for each $\lambda$ choices. The table below shows that our choice of $\lambda=20.0$ presents stable results and the performance among different $\lambda$ choices is relatively robust, without dramatic performance drops.
>
> | $\lambda$ | DVTS w/ REAR (Mean ± Std %) | BoN w/ REAR (Mean ± Std %) |
> | :-------- | :-------------------------- | :------------------------- |
> | **1.0**   | 74.66 ± 0.75                | 76.02 ± 0.64               |
> | **3.0**   | 75.96 ± 0.42                | 77.45 ± 0.79               |
> | **10.0**  | 78.08 ± 0.67                | 77.57 ± 0.46               |
> | **20.0**  | **78.62** ± 0.67            | **78.42** ± 0.63           |
> | **50.0**  | 77.92 ± 0.80                | 78.05 ± 1.00               |
> | **100.0** | 77.54 ± 0.54                | 77.85 ± 1.23               |
> | **200.0** | 77.98 ± 0.94                | 78.15 ± 1.05               |
>
> **About Automated Selection:** Since REAR is a training-free method without much computational cost, there will not be significant efforts to find the best $\lambda$ value. It is expected to develop heuristic methods or derive the best $\lambda$ with prior knowledge, which are **already indicated in our statement of future work**. In the revised version, we provide more details on this direction.
>
> > If user preferences are not expressed in words (for example, are only implicit, behavioral, or external), the REAR method as proposed would not function without modification.
>
> Actually we have considered different forms of preference expressions. In our selected benchmarks, the preference are in both **explicit and implicit** forms. Taking the Prefeval benchmark as an example:
>
> 1. PrefEval **explicit preference** tasks use explicit preference sentences.
> 2. PrefEval **implicit choice** tasks use two-turn conversations which implicitly contain the preference information.
> 3. PrefEval **implicit preference** tasks use multi-turn conversations to express the preference in the context.
>
> We show that our representation of preference information can be well expanded to not a simple system message but complex multi-turn conversations implicitly. We appreciate the idea that the preference can be further extended to non-textual modalities like images and behaviors. It is potential to adopt a multimodal language model to handle this scenario. We revise the paper to include future directions on that.
>
> > Heavy reliance on LLM-based evaluation for some tasks raises concerns about evaluation robustness and objectivity.
>
> We share the concern about LLM-as-a-judge. To mitigate this, we also included the **PrefEval Implicit Choice** task, which is evaluated via **Accuracy (Ground Truth)**, not an LLM judge. REAR achieves **78.6% accuracy** vs. 71.5% (Greedy) on this objective metric. Meanwhile, the consistency between the Objective Accuracy results and the Subjective LLM-Judge results strongly suggests that our performance gains are real and not merely an artifact of evaluation bias.

---

> > ### Author Response · Authors · 2025-11-25
> >
> > > It it insufficient to use only the “helpfulness” score to measure the general response quality in the Section "Analysis on Generated Responses", and also "helpfulness" can be one of the preferences to consider.
> >
> > We apologize for the confusion in the helpfulness score to measure the general response quality. To avoid designing controversial metrics to measure them, we directly borrow multiple rubrics judged by the PrefEval benchmark in Figure 3, including helpfulness, violation of preference, and preference acknowledgement.
> >
> > **Why is "helpfulness" a general metric?** That is because the official implementation of evaluating helpfulness does not require the preference information as a part of input prompt, merely evaluating the general helpfulness of the response.
> >
> > **Other metrics of general response quality.** We acknowledge that there are other metrics to measure general response qualities like safety-related metrics. However, the PrefEval benchmark does not include them as rubrics to compute the final score. They are also not related to our task of preference alignment. As a result, we use helpfulness as the metric of general responses since it is more suitable for this benchmark.
> >
> > ---
> >
> > We hope these clarifications address your concerns. Please let us know if you have any further questions.

---

### Official Review · Reviewer_GZ9K · 2025-10-24

**Soundness:** 2
**Presentation:** 3
**Contribution:** 2
**Rating:** 2
**Confidence:** 4

**Summary:**

This paper proposes REAR, a test-time method for aligning large language models with user preferences without further training. The key idea is to decompose the implicit reward into question-related and preference-related components and to rescale the preference term at inference time. The authors show that REAR can be computed as a linear combination of log-probabilities and integrated into existing test-time scaling (TTS) algorithms such as best-of-N sampling and diverse verifier tree search (DVTS). Experiments on several preference-alignment benchmarks (PrefEval, Multifaceted, PingPong) demonstrate modest gains compared to existing TTS and test-time alignment baselines.

**Strengths:**

- The paper is well written and clearly structured, with theoretical derivations and reasonable experimental design.

- The proposed formulation is efficient and lightweight, requiring no extra training or reward models.

**Weaknesses:**

- The idea of using log-probabilities or policy scores for implicit reward shaping at test time has been explored in prior work. The contribution mainly reformulates known ideas in a slightly different analytical framing. The proposed reward decomposition essentially feeds the reward model (or policy probability) with different segments of the same input (question vs. question + preference) to derive a rescaled score. This approach, while intuitive, lacks genuine novelty or theoretical depth.

- In the second paragraph of the introduction part, the authors state that "However, existing TTS research has predominantly focused on domains such as mathematics and coding". However, test-time alignment has been long studied, even before the prevalence of TTS, including papers such as:

1. Args: Alignment as reward-guided search.

2. Inference-time language model alignment via integrated value guidance.

3. Fast Best-of-N Decoding via Speculative Rejection.


- Performance gains over baselines are modest, often within small margins and without strong qualitative differentiation.

- The method remains heuristic: while mathematically presented as a decomposition, it does not provide clear theoretical or empirical evidence that the “preference” and “question” components can be distinctly separated in practice.

**Questions:**

See weaknesses.

---

> ### Author Response · Authors · 2025-11-25
>
> > The idea of using log-probabilities or policy scores for implicit reward shaping at test time has been explored in prior work. The contribution mainly reformulates known ideas in a slightly different analytical framing. The proposed reward decomposition essentially feeds the reward model (or policy probability) with different segments of the same input (question vs. question + preference) to derive a rescaled score. This approach, while intuitive, lacks genuine novelty or theoretical depth.
>
> We argue the novelty lies in how these segments are used. Prior "Contrastive Decoding" methods use these logits to modify the sampling distribution (next-token prediction).
>
> REAR’s contribution is deriving a scalar reward token ($r_{REAR}$) from these logits. This shift from "decoding heuristic" to "defined reward" is what allows us to integrate with Tree Search (DVTS).
>
> - Previous logit-manipulation methods (like Contrastive Decoding) cannot inherently perform lookahead search.
> - By formalizing the logit-difference as a *reward* in an MDP (Lemma 3.1), we unlock the use of sophisticated search algorithms (like DVTS) for **subjective preference alignment**—a domain previously excluded from Test-Time Scaling because it lacks ground-truth verifiers (unlike Math/Code).
>
> > The idea of using log-probabilities or policy scores for implicit reward shaping at test time has been explored in prior work including ARGS, Inference-time language model alignment via integrated value guidance, Fast Best-of-N Decoding via Speculative Rejection
>
> We thank the reviewer for these valuable references and will include them in our related work section. However, we respectfully submit that **REAR differs fundamentally from these approaches in two critical ways:**
>
> 1. **Training-Free vs. Training-Required:** Methods like Inference-time value guidance (*IVG*) require training a value function or a reward model. *ARGS* (Alignment as Reward-Guided Search) also relies on an external reward model to guide the search. In contrast, REAR extracts a dense reward signal directly from the base model's own probabilities. This is a crucial distinction for scalability, as it avoids the cost of training and deploying separate reward/value models.
> 2. **Simple Decoding vs. Search-based:** While *Fast Best-of-N* focuses on efficient sampling, it operates under the standard Best-of-N paradigm. Our contribution goes further by formulating the self-reward in a way that enables more complex search-based TTS methods like **DVTS**. This allows us to bring step-by-step search to subjective preference tasks, which is a novel application. In our experiments, we also have compared our methods with multiple variants of Best-of-N approaches, proving the effectiveness of REAR.
>
> > Performance gains over baselines are modest, often within small margins and without strong qualitative differentiation.
>
> We disagree with this point. As summarized in the following Table, our performance gain averaged from all benchmarks are significant. Even compared with BoN using external RM, we can still acquire sufficient gains without using additional models.
>
> | Baseline         | Greedy | Amulet | LA     | GenRM  | External RM |
> | ---------------- | ------ | ------ | ------ | ------ | ----------- |
> | Performance Gain | +18.1% | +15.9% | +14.9% | +13.8% | +4.5%       |
>
> > The method remains heuristic: while mathematically presented as a decomposition, it does not provide clear theoretical or empirical evidence that the “preference” and “question” components can be distinctly separated in practice.
>
> We note that this separation is not a heuristic assumption but a theoretical consequence of the Maximum Entropy RL framework used to model LLM generation (Lemma 3.1 & 3.2). In MaxEnt RL, the optimal policy distribution is directly proportional to the soft Q-function under the reward $r'(s, a)$. This fundamental property allows us to mathematically recover the implicit rewards driving the model's behavior.
>
> As the base model optimizes a composite reward: $r'(s, a) = r'(s^q, a) + \alpha \cdot r_p(s, a)$, and the question-only model $\pi(a|s^q)$ optimizes $r'(s^q, a)$, subtraction on its inner soft-Q function helps us **cancel out the question-related terms**, thus **isolating the preference reward $r_p(s, a)$**. Therefore, we can compose the realignment reward from a reshaped reward $\hat{r}_\text{REAR}$. This proves that our method does not **merely balance logits heuristically**. Instead, it extracts a mathematically **rigorous reward signal representing pure preference realignment**. This validity allows us to use the signal not just for sampling, but as a value function in lookahead search (DVTS).
>
> ---
>
> We hope these clarifications address your concerns. Please let us know if you have any further questions.

---

### Official Review · Reviewer_Uh7X · 2025-10-31

**Soundness:** 2
**Presentation:** 1
**Contribution:** 2
**Rating:** 2
**Confidence:** 4

**Summary:**

This paper proposes an idea of reward decomposition that is used in test-time inference for the personalized preference task. The idea is simple, decompose the whole reward into question-related reward and preference-related reward. Then, the authors use the policy probability $\pi(a|s)$ as a proxy of the reward $r(s,a)$.

Based on these reward decomposition, the authors use them in two test-time algorithms, Best-of-N and DVTS. Experiments demonstrate the effectiveness/efficiency of their reward design.

**Strengths:**

The reward decomposition strategy is simple and easy to understand, while the effectiveness is demonstrated by the experimental results.

**Weaknesses:**

1. The authors tried to use some mathematical derivation to show the depth of their approach, however, I feel these components were not well stated. For example: From (5) to (7), it feels like nothing substantial is explained. Lemma 3.1 also appears quite abrupt — it introduces a seemingly fancy formula, but its meaning is unclear, and in the end, everything just circles back to (7). Note that Lemma 3.1 is merely an intermediate result and does not offer any theoretical guarantee. It only shows that your policy is an optimizer of a certain expression, making the interpretation of this relation crucial. However, I don't find it particularly insightful or helpful for understanding your algorithm’s design. In fact, it ultimately leads to equation (7), which is equivalent to (5).

2. What truly matters is Lemma 3.2, as it informs the reader how to compute the reward. However, the authors merely state that the reward can somehow be replaced by the policy probability. I believe this is a critical step in the algorithm’s design, yet the paper provides no intuition or explanation for this substitution. Although proofs are included in the appendix, they also fail to offer any meaningful intuition.

2. The presentation could be improved for better clarity and coherence. For example:
 (a) What is the REAR score, and how is it defined? (I can roughly infer its meaning, but the paper should state it explicitly.)
 (b) What is $\hat{r}_{REAR}$ in Lemma 3.2?
 (c) In Proposition 2.1 and several other places, you should distinguish between the symbols '=' (equality) and ':= / \triangleq' (definition). This is especially important when your notation deviates from standard conventions. For instance, in Equation (3), your Q function is not the one that I learned. If you want to define a new Q function (or the soft Q-function, as you called), you should use a definition symbol rather than the equality sign.

3. After analyzing the paper, I find the novelty is quite limited. It decompose the reward into two reward terms, then replaces the two reward terms by two policy probability terms, and use it as guidance in two common test-time inference methods. The mathematical derivations make little sense and sometimes disrupt the reading flow of the paper.

**Questions:**

see weakness.

---

> ### Author Response · Authors · 2025-11-25
>
> #### The Intuition Behind Lemma 3.2
>
> > The paper provides no intuition or explanation for this substitution of replacing reward with policy probability.
>
> We agree that this intuition is central to the paper and should have been stated more explicitly in the main text. The substitution is grounded in **Maximum Entropy Reinforcement Learning (MaxEnt RL)**, which is stated in Section 3.2. The intuition is that the optimal policy $\pi^*(a|s)$ takes the form of a Boltzmann distribution proportional to the exponential of the soft Q-function2 $$\pi^*(a|s) \propto \exp\left(\frac{Q^*(s,a)}{\beta}\right)$$. Since the Q-function represents the long-term cumulative reward, the log-probability of the policy is directly proportional to the reward, plus value function terms.
>
> **Why this matters for REAR:** Lemma 3.2 utilizes this relationship to express the realignment reward. Since we have the pre-trained policy $\pi$, we can mathematically recover the implicit reward it is optimizing. By taking the log-probabilities of the model's own output, we gain access to a dense, token-level reward signal without needing to train a separate reward model. We will add this explicit intuitive explanation to Section 3.1 in the revision.
>
> #### The Purpose of Lemma 3.1 and the Derivation from Eq. (5) to (7)
>
> > From (5) to (7), it feels like nothing substantial is explained. Lemma 3.1 also appears quite abrupt — it introduces a seemingly fancy formula, but its meaning is unclear, and in the end, everything just circles back to (7). Note that Lemma 3.1 is merely an intermediate result and does not offer any theoretical guarantee.
>
> We clarify that Lemma 3.1 is not merely circular. It provides the **theoretical justification for why simple logit manipulation equates to "Realignment."**
>
> - **Without Lemma 3.1:** Combining log-probs (Equation 9) looks like a heuristic hack, essentially contrastive decoding.
> - **With Lemma 3.1:** We prove that the base model $\pi$ is *already* the optimal solution to an objective maximizing a preference reward $r_p$ subject to a KL constraint toward the question-only policy $\pi_q$. In this case, we want to realign the policy since the KL constraint can be too strong or too weak in real tasks.
>
> Lemma 3.1 identifies that the pre-trained model implicitly solves a constrained optimization problem. This allows us to define Realignment not as changing the logits, but as **changing the Lagrange multiplier ($\lambda$)** of that optimization problem. This theoretical step is necessary to justify using the resulting score in **Tree Search (DVTS)**, which requires a valid reward function, rather than just a decoding heuristic.
>
> #### Presentation and Notation
>
> > What is $\hat{r}_\text{REAR}$ in Lemma 3.2?
>
> As we stated in Lemma 3.2, $\hat{r}_\text{REAR}$ is a proxy reward of the original REAR $r_\text{REAR}$ reward. In Appendix A.3, we show that $\hat{r}_\text{REAR}$ is a potential-based reward shaping of REAR. As a result, it will keep policy optimality with REAR, which gives us the insight to use $\hat{r}_\text{REAR}$ for response selection.
>
> > What is the REAR score, and how is it defined?
>
> **REAR Score:** The REAR score $S(s,a)$ is derived from $\hat{r}_\text{REAR}$ in Equation (8). We only introduce $\lambda$ to replace previous multiple coefficients for a simple expression. We denote it as a score function since in our test-time scaling method, we directly compute the REAR score of each response or response segment.
>
> > In Proposition 2.1 and several other places, you should distinguish between the symbols '=' (equality) and ':= / \triangleq' (definition). This is especially important when your notation deviates from standard conventions. For instance, in Equation (3), your Q function is not the one that I learned. If you want to define a new Q function (or the soft Q-function, as you called), you should use a definition symbol rather than the equality sign.
>
> We will rigorously distinguish between equality ($=$) and definition ($:=$), particularly in Proposition 2.1. Regarding the Q-function in Eq. (3), we want to recall that the soft Q-function is not a new definition proposed by us and we do not include the standard Q-function through this work. we will clarify that this refers specifically to the **Soft Q-function** used in MaxEnt RL (incorporating the entropy term), which differs from the standard Q-function in classical RL.

---

> > ### Author Response · Authors · 2025-11-25
> >
> > #### Novelty
> >
> > > It decompose the reward into two reward terms, then replaces the two reward terms by two policy probability terms, and use it as guidance in two common test-time inference methods. The mathematical derivations make little sense and sometimes disrupt the reading flow of the paper.
> >
> > First, as answered in previous questions, our theory part has its meanings to prove that our test-time method can **maximize a realigned reward objective** rather than **simply assembling the log-probabilities**. On the other hand, we think the simplicity of the final REAR score is a strength of our method, not a limitation. Compared to related methods, REAR has the following advantages:
> >
> > - **Scalable Subjective Alignment:** Prior Test-Time Scaling (TTS) work focuses on verifiable tasks (Math/Code) because ground-truth rewards exist. For subjective preference alignment, no such ground truth exists. REAR provides a novel mechanism to **extract** a dense, token-level reward from the model itself, enabling applying sophisticated search algorithms (like DVTS) to these tasks.
> > - **Efficiency:** Unlike methods that require training or deploying external reward models (which also double the memory cost) or classifiers, REAR requires **no training** and minimal compute overhead, yet still outperforms external Reward Model baselines (Table 1).
> >
> > We believe bridging the gap between subjective preference alignment and tree-search-based TTS without external supervision is a significant contribution to the field.
> >
> > ---
> >
> > We hope these clarifications address your concerns. Please let us know if you have any further questions.

---

### Official Review · Reviewer_V7V3 · 2025-10-31

**Soundness:** 2
**Presentation:** 2
**Contribution:** 2
**Rating:** 4
**Confidence:** 3

**Summary:**

This paper presents a test-time scaling method for preference alignment in LLMs, in case of non-verifiable rewards. This paper makes the assumption that preference is specified in-context. The authors decompose rewards into question-related and preference-related components, then derive a score based on policy probabilities that can be integrated with best-of-N sampling and DVTS. While the core idea has merit, the paper suffers from significant theoretical gaps, questionable experimental design, and overclaimed contributions.

**Strengths:**

1. The "realignment" framing is intuitive: base models have implicit preferences from training that may not match specific user needs.

2. The paper correctly identifies that test-time scaling (TTS) has been limited to verifiable domains (math, coding) and extending it to subjective preference alignment is a worthwhile research direction.

**Weaknesses:**

1. What is α? Is it:
  - Task specific constant
  - A property of how the model was trained?

2. This paper would have been much easier to understand if it were presented as "Test-Time Preference Alignment via Policy Interpolation". Overcomplicating a simple method doesn't add value to the paper.

3. Experimental Design Lacks Statistical Rigor. Statistical significance of the results is not reported.

**Questions:**

Please see weaknesses.

---

> ### Author Response · Authors · 2025-11-25
>
> > What is α? Is it task specific or a property of training?
>
> α is a task-level coefficient that controls the relative emphasis between the **question-related reward** and the **preference-related reward**. $\alpha$ is **a property of the pretrained base model**. In our derivation (Equation 5), we posit that the implicit reward $r'(s, a)$ optimized by the pretrained model is a linear combination of a question-answering reward $r'(s^q, a)$ and a preference-adherence reward $r_p(s, a)$.
>
> * **$\alpha$ represents the inherent weight** the base model assigns to preference adherence resulting from its original training data distribution.
> * **$\hat{\alpha}$ represents our target weight** for the realignment task.
>
> Crucially, our final derived score (Equation 9) depends only on the ratio $\lambda = \hat{\alpha}/\alpha$. Therefore, we do not need to know the exact value of the inherent $\alpha$. By tuning the hyperparameter $\lambda$, we effectively scale the importance of the preference reward relative to the base model’s original behavior.
>
> > This paper would have been much easier to understand if it were presented as "Test-Time Preference Alignment via Policy Interpolation".
>
> The final score formulation of the REAR score $S(s, a)$ mathematically resembles policy interpolation. However, we argue that the **Reward Decomposition framing is essential, not an overcomplication**, for two key reasons:
>
> 1. **Theoretical Grounding for Test-Time Scaling:** If we viewed this merely as "Policy Interpolation," the natural method would be to simply sample from the interpolated distribution (like Classifier-Free Guidance). However, our goal is **Test-Time Scaling**, aiming at maximizing the realignment reward ($r_{REAR}$) under the MDP formulation.
> 2. **Optimality Proof:** Lemma 3.1 proves that the base model is *already* the optimal policy for a specific reward decomposition. This proves that our method isn't just a heuristic modification of logits, but a mathematically principled way to shift the reward objective while maintaining the KL-regularization structure of the base LLM.
>
> > Experimental Design Lacks Statistical Rigor. Statistical significance of the results is not reported.
>
> We agree that statistical significance is important. We do not conduct multiple runs mainly due to the cost issues under LLM evaluations, which also follows our compared baselines [1,2]. However, we acknowledge that including experimental results with different seeds can better show the significance of our method. As a result, we run **five** experiments for the PrefEval implicit choice benchmark, which adopts rule-based evaluation.
>
> | Method              | Mean      | Std Dev     | 95% CI (±)  |
> | :------------------ | :-------- | :---------- | :---------- |
> | DVTS w/ REAR (Ours) | **78.62** | 0.67        | 0.83        |
> | BoN w/ REAR (Ours)  | 78.42     | 0.63        | 0.67        |
> | External RM         | 77.68     | 0.74        | 0.78        |
> | GenRM               | 76.28     | 1.22        | 1.51        |
>
> Furthermore, we want to highlight that the performance gaps we observe are **substantial and consistent**. As summarized in the following Table, our performance gain averaged from all benchmarks are significant. Even compared with BoN using external RM, we can still acquire sufficient gains without using additional models.
>
> | Baseline         | Greedy | Amulet | LA     | GenRM  | External RM |
> | ---------------- | ------ | ------ | ------ | ------ | ----------- |
> | Performance Gain | +18.1% | +15.9% | +14.9% | +13.8% | +4.5%       |
>
> ---
>
> We hope these clarifications address your concerns. Please let us know if you have any further questions.
>
> [1] Amulet: ReAlignment During Test Time for Personalized Preference Adaptation of LLMs.
>
> [2] Linear Alignment: A Closed-form Solution for Aligning Human Preferences without Tuning and Feedback.

---

### Meta-Review · Area_Chair_UhkW · 2026-01-11

**Summary:**

The paper proposes **REAR**, a training-free test-time realignment method for LLMs on *subjective preference alignment* tasks. The central modeling choice is to treat generation as optimizing an implicit composite reward that can be **decomposed** into (i) a *question-related* component and (ii) a *preference-related* component. REAR aims to **rescale** the preference-related component at test time while preserving the question-related component. The resulting REAR score is presented as a **linear combination of log-probabilities** from policies conditioned on different contexts (question-only vs. question+preference), and is used as a reward signal that can plug into **Best-of-N** and **DVTS** test-time scaling. Experiments on preference/role-playing benchmarks (e.g., PrefEval, Multifaceted, PingPong) show improved preference adherence relative to greedy decoding and several test-time alignment baselines, with some results also reported on a rule-based/ground-truth setting (PrefEval implicit choice) and added multi-run statistics in rebuttal for that benchmark.

### Strengths
- **Training-free, plug-and-play** approach that avoids external reward models and additional finetuning, and is designed to integrate into existing TTS methods (BoN, DVTS).
- **Empirical improvements** reported across multiple benchmarks; rebuttal adds **multi-run results** (mean/std/CI) for PrefEval implicit choice and reports robustness trends over the key hyperparameter (λ).

### Weaknesses
- **Novelty concerns**: reviewers argue the core mechanism resembles prior logit/probability-based guidance or contrastive decoding-style ideas, and the main contribution may be a reframing rather than a substantially new method.
- **Evaluation concerns**: some benchmarks rely on LLM-as-a-judge; while the paper includes at least one objective benchmark and rebuttal emphasizes consistency, the broader evaluation robustness remains somewhat limited.
- **Hyperparameter sensitivity**: performance depends on λ; rebuttal provides evidence of empirical robustness but no principled automated selection method.

**Reviewer Concerns:**

### Addressed
- **V7V3 (α meaning; statistical rigor):** Authors clarified α as a task-level coefficient / inherent base-model preference weight and emphasized only the ratio matters; they also added **5-run statistics** for PrefEval implicit choice and provided average gains.
- **Uh7X (intuition for Lemma 3.2; purpose of Lemma 3.1; notation/presentation):** Rebuttal adds a direct MaxEnt RL intuition connecting log-probabilities and (soft) Q/reward, and explains Lemma 3.1 as a justification for interpreting the procedure as changing a Lagrange multiplier rather than ad-hoc logit mixing; commits to fixing notation and definitions in revision.
- **NCzj (λ robustness; evaluation reliance):** Rebuttal provides a λ sweep with multiple runs and discusses scenario-dependent trends; highlights inclusion of an objective (ground-truth) benchmark and argues consistency between objective and LLM-judge results.

### Still outstanding
- **Core novelty/positioning (Uh7X, GZ9K):** Even with clarified framing (reward vs decoding heuristic), it remains contested whether the contribution is materially beyond existing logprob/logit-guidance approaches, especially given the simplicity of the final scoring rule.
- **Evaluation breadth/rigor (V7V3, NCzj, GZ9K):** Added statistics are limited to one benchmark and do not fully address concerns about the reliance on LLM-judging elsewhere.

**Reviewer Scores:**

Likely no changes.

---

### Decision · Program_Chairs · 2026-01-26

Reject